# Software evolution: the lifetime of fine-grained elements

Diomidis Spinellis[1,2], Panos Louridas[1] and Maria Kechagia[3]

[1] Department of Management Science and Technology, Athens University of Economics and Business, Athens, Greece
[2] Department of Software Technology, Delft University of Technology, Delft, The Netherlands
[3] Department of Computer Science, University College London, London, UK

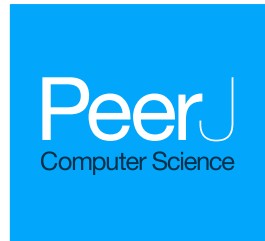

## ABSTRACT

A model regarding the lifetime of individual source code lines or tokens can estimate maintenance effort, guide preventive maintenance, and, more broadly, identify factors that can improve the efficiency of software development. We present methods and tools that allow tracking of each line's or token's birth and death. Through them, we analyze 3.3 billion source code element lifetime events in 89 revision control repositories. Statistical analysis shows that code lines are durable, with a median lifespan of about 2.4 years, and that young lines are more likely to be modified or deleted, following a Weibull distribution with the associated hazard rate decreasing over time. This behavior appears to be independent from specific characteristics of lines or tokens, as we could not determine factors that influence significantly their longevity across projects. The programing language, and developer tenure and experience were not found to be significantly correlated with line or token longevity, while project size and project age showed only a slight correlation.

## INTRODUCTION

Although there is a significant body of work regarding the macroscopic characteristics (*González-Barahona et al., 2009*) and even laws (*Lehman, 1980*) of software evolution (*Herraiz et al., 2013*), much less is known about how software evolves at the microscopic scale, namely at the level of lines, statements, expressions, and individual tokens. A study of such details, apart from its self-supporting merits as curiosity-driven empirical research, can derive results that can in the future be used for improving software development processes (*Humphrey, 1989*, p. 3), architecting software systems (*Barnes, Pandey & Garlan, 2013*; *Breivold, Crnkovic & Larsson, 2012*), developing machine learning algorithms (*Allamanis et al., 2018*; *Alon et al., 2019*), organizing software development teams (*Rodríguez et al., 2012*), estimating maintenance effort (*Albrecht & Gaffney, 1983*; *Atkins et al., 2002*; *Zimmermann et al., 2005*), designing new features for configuration management systems (*White et al., 2015*; *Jiang, Armaly & McMillan, 2017*), locating software faults (*Cotroneo, Natella & Pietrantuono, 2013*; *Giger, Pinzger & Gall, 2011*; *Kechagia et al., 2019*; *Salfner, Lenk & Malek, 2010*), guiding probabilistic programing (*Gordon et al., 2014*), and enhancing programing languages (*Vallée-Rai et al., 2010*). Here we report on methods, tools, and the results we obtained by studying the lifetime of

Corresponding author
Diomidis Spinellis, dds@aueb.gr

unmodified code lines and tokens in 89 revision control repositories over 3.3 billion source code element lifetime events.

At the point where the rubber hits the road, software consists of code lines. Their number has been extensively studied to gain insights on topics ranging from development effort (*Albrecht & Gaffney, 1983*; *Gousios, Kalliamvakou & Spinellis, 2008*; *Lind & Vairavan, 1989*) and quality (*Buse & Weimer, 2008*; *Kan, 2002*; *Stamelos et al., 2002*; *Zhang, 2009*) to software growth (*Van Genuchten & Hatton, 2013*; *Godfrey & Tu, 2000*; *Hatton, Spinellis & Van Genuchten, 2017*; *Herraiz, González-Barahona & Robles, 2007*). This work contributes to the study of software evolution by looking quantitatively, not at changes in the *number* of code lines, but how and why individual lines or tokens change over the software's lifetime.

First, consider how long a line of code survives in its initial form. As software evolves over time, some lines are added, others are deleted, and existing ones are modified. From the time that a line enters the code base of a project, for how long does it live, that is, for how long does it remain there unchanged? Are lines of code more of a durable asset that will be around for the long time, or are they more like perishable assets, that will only remain for a short time? How is their lifetime related to factors such as a system's size or the employed programing language?

A process model of aging can be further elaborated through quantitative characteristics. These include the mathematical function that determines when existing lines are likely to "die". We define as the line's death its deletion or the modification of its non-whitespace elements, and further examine the validity of this construct by also looking at the lifetimes of individual tokens. In functions that are used to characterize decay processes, their characteristic unit is often expressed through the measure of *median lifespan*: $t_{1/2}$. If a line $i$ is added at time $t_{i,1}$ and is changed or disappears at time $t_{i,2}$, its lifespan is $t_{i,2} - t_{i,1}$. The median lifespan, over all lines of a project, is the median value of all line lifespans, that is, the median of $t_{i,2} - t_{i,1}$ for all $i$.

Now take an added line of code. When will this code be changed or be entirely removed and how does its age factor into this question? One can imagine three possibilities. The first, a *high infant mortality* scenario, in which new lines of code are often changed as developers fix newly-introduced faults and refactor the code. The second, a *senescence* scenario, has code lines become outdated and less useful as they age and therefore face increasing chances of being replaced. The third, *stochastic* scenario, has lines getting replaced mostly due to other factors through what appears to be a random process with regard to their age. In practice, it is likely that all three scenarios play a role, but it is still possible that one of them dominates the aging process.

Finally, consider some reasons for which a line may change. These include misunderstood requirements, a fault in the code, or changes cascading from other work. While these are typically qualitatively analyzed, one can also examine the factors associated with them, such as the line's complexity, its developer's seniority, the project's size, or the employed programing language.

Apart from its theoretical importance, a model of code aging at the level of code lines is useful in several ways. Many potential applications are listed in the opening paragraph;

here are two concrete examples. First, the model can inform managers where to direct maintenance effort, for example to reduce the acquired technical debt (*Kruchten, Nord & Ozkaya, 2012*) or address newly-discovered security vulnerabilities (*Ozment & Schechter, 2006*; *Penta, Cerulo & Aversano, 2009*; *Shin et al., 2010*). Under the infant mortality scenario old lines are likely to remain in a project for ages, so they should periodically receive some love and care to keep them up to date with modern practices. In contrast, under the senescence scenario these will gradually fade away, so effort invested in maintaining them may be wasted. Second, the function expressing code age and its coefficients for a specific project can be used to guide maintenance effort estimation. This is important because humans are often biased when estimating development effort (*Løhre & Jørgensen, 2016*). Simplistically, effort is often measured in terms of code lines (*Albrecht & Gaffney, 1983*; *Gousios, Kalliamvakou & Spinellis, 2008*; *Lind & Vairavan, 1989*). Therefore, if, given the age of existing lines, we can estimate how many of the project's lines are likely to change in the next year, this, together with the project's estimated code growth rate (*Hatton, Spinellis & Van Genuchten, 2017*), can roughly determine the required development effort. More broadly and importantly, given that code lines require effort to change, identifying and controlling factors that promote longer-living lines—for instance through better abstraction mechanisms—can direct improvements in software development efficiency.

The contributions of this article are the development of an efficient method and tools that allow the tracking of the birth and death of individual source code lines and tokens over periods that can span decades and the empirical analysis of 3.3 billion source code element lifetime events to answer the following research questions.

**RQ1** For how long does a line of code or token live? The answer to this question determines whether code elements are durable or perishable.

**RQ2** How is a line's or token's age related to the probability of its change or deletion? The answer tells us whether younger code elements are more vulnerable to change and deletion (infant mortality), or whether older ones are more frail (senescence), or whether there are no age-related hazards.

**RQ3** What other product or process factors may influence a line's or a token's survival? We investigate this question along the following dimensions.

**RQ3a** The line's characteristics, which may reveal change-prone programing constructs or drivers of change.

**RQ3b** The different token types, which may affect the lifetime of the tokens.

**RQ3c** The committer's experience and tenure; one might expect senior developers to write more stable code.

**RQ3d** The project's size, which might lend it inertia against change.

**RQ3e** The employed programing language, demonstrating whether some programing languages lend themselves for writing more stable (or, alternatively, flexible) code.

## METHODS

We studied code aging at the level of individual source code lines by selecting a number of suitable revision control repositories to study, coming up with a way to handle merges of

development branches, constructing a tool that can track the lifetime of individual lines across successive software releases, creating a process and tools to also study the lifetime of individual tokens, choosing the statistical methods that best suited the domain, and applying them to the collected data.

As recommended by *Ince, Hatton & Graham-Cumming (2012)*, the source code and data associated with our results are openly available online.[1]

[1] Data: https://doi.org/10.5281/zenodo.4319986 (3.5 GB compressed, 69 GB uncompressed); source code: https://doi.org/10.5281/zenodo.4319993.

## Material selection

We ran our study on open source software repositories due to their liberal availability and the fact that this simplifies the replication of our findings. We selected the revision control repositories to study based on five objective criteria consistent with our research goals.

### GitHub hosting

We only selected projects whose history is available on GitHub. This decision simplified the methods we used to select the projects and to traverse a project's revisions. The choice to use only GitHub-hosted repositories is not as restrictive as it sounds, because nowadays even projects that use other version control systems and hosting often make a (typically read-only) Git version of their repository available on GitHub.

### Longevity

We selected projects with at least ten years of commit data in order to obtain enough samples for statistical analysis.

### Active development

The code in the repository had to be actively developed as determined by code commits. Obviously, code in dormant projects does not exhibit aging processes and cannot be usefully modeled. To examine projects that are actively developed we calculated the number of weeks over the project's lifetime in which at least one commit had occurred. We then selected projects in which commits occurred in at least 85% of the weeks to take into account vacation time. We examined activity at weekly granularity, because some open source developers may only work over weekends.

### Popularity

We selected projects having at least 100 GitHub "stars". Results from popular projects are likely to be relevant to more people than those from obscure ones. Studying code aging in small test projects or student exercises is less likely to yield results of industrial relevance.

### Programming language

To study source code evolution at the level of individual tokens as well as lines, we only selected projects whose main programing language, as reported in GHTorrent (*Gousios & Spinellis, 2012*), is supported by the tokenizer we used (*Spinellis, 2019*). These languages were selected based on their popularity among the projects selected using

**Table 1 Descriptive statistics of the analyzed repositories and aggregate totals.**

| Metric | Min | Max | Median | Mean | σ | Total |
|---|---|---|---|---|---|---|
| All files | 384 | 110,737 | 2,786 | 7,986 | 17,522 | 710,798 |
| Analyzed Source Code Files | 165 | 73,655 | 1,291 | 4,175 | 11,177 | 371,566 |
| Analyzed source code lines (thousands) | 3 | 15,699 | 298 | 935 | 2,083 | 83,204 |
| Analyzed source code tokens (thousands) | 0 | 72,480 | 1,423 | 4,537 | 9,793 | 403,804 |
| Committers | 29 | 2,211 | 247 | 397 | 407 | 35,374 |
| GHTorrent project duration (Years) | 11 | 29 | 12 | 14 | 4 | |
| Analyzed branch duration (Years) | 6 | 41 | 13 | 15 | 5 | |
| All commits (thousands) | 6 | 299 | 19 | 39 | 51 | 3,508 |
| Analyzed commits (thousands) | 0 | 192 | 11 | 24 | 36 | 2,150 |
| Line deaths (thousands) | 1 | 28,680 | 827 | 2,326 | 4,482 | 207,022 |
| Token deaths (thousands) | 0 | 157,530 | 3,266 | 9,954 | 21,516 | 885,935 |
| Project stars | 117 | 34,193 | 878 | 2,207 | 4,289 | |
| Commit density (weekly %) | 85 | 100 | 93 | 93 | 5 | |

the other criteria, and cover 76% of the repositories initially selected. The languages processed are C, C#, Java, PHP, and Python.

We performed the project selection through analytical processing of the GHTorrent data set (January 2018 version) based on the process described by *Gousios & Spinellis (2017)*. The code information was obtained by downloading and processing each selected revision of the corresponding repository. Other than the stated selection criteria, we did not perform any other manual adjustments to add popular projects or exclude obscure ones. We ensured that our data set did not include duplicated projects by pairing it with a dataset for GitHub repository deduplication (*Spinellis, Kotti & Mockus, 2020*). From the one duplicate and two triplicate sets we thus found we retained the repositories with the longer commit history. Specifically, we chose github.com/droolsjbpm/drools over kiegroup/drools and droolsjbpm/guvnor, lede-project/source over openwrt-mirror/openwrt and openwrt/packages, and doctrine/doctrine2 over doctrine/dbal. In total we analyzed 89 projects, comprising at the end of the studied period 372 thousands of source code files, 83 millions of source code lines, 404 millions of source code tokens, and 2.2 millions of examined commits. In terms of source code lines, we analyzed 497 million code line lifetime events: the appearance of 290 and the demise of 207 millions of source code lines. In terms of individual source code tokens, we analyzed 2.2 billion code token lifetime events: the appearance of 1,290 and the demise of 886 millions of source code tokens. Key metrics of the projects we analyzed are listed in Table 1.

## History simplification

Software development with distributed version control repositories is characterized by the frequent generation of feature development branches and their subsequent merging into a more mainstream trunk (*German, Adams & Hassan, 2016*). For example, the repositories we analyzed contained a total of 243 thousand merges, or about three

thousand per project. As we detail below, merges confuse and complicate the processing of history and therefore required devising a scheme to deal with them.

The confusion arises from the fact that a topological ordering implied by the directed acyclic graph structure of a project's commit history will lose code changes or their time ordering. (A topological ordering of a directed graph is a linear ordering of its vertices such that for every directed edge $ab$ from vertex $a$ to vertex $b$, the linear ordering has $a$ appear before $b$.) Applying the typically-used three-way merge algorithm will result in the loss of code modifications, because the common ancestor will no longer be correctly represented.

The complexity of merges has to do with how changes listed at the point of a merge can be automatically processed to obtain the code after it. In experiments we performed with diverse Git output options we found that the output of *git log* and also *git format-patch* was not a reliable way to reconstruct the state of a project from its history (*Spinellis, 2016a*). Consequently, the output could also not be used to track the lifetime of individual lines. Although we did not look for the root cause of these problems in depth, we only encountered them when working with merges, which leads us to believe that they were indeed caused by merges. Using Git's combined diff format for merges is also unlikely to help, because, according to Git's *git diff* documentation "Combined diff format was created for review of merge commit changes, and was not meant for apply". And if dealing with binary merges was not bad enough, handling $n$-way merges, such as those handled by Git's octopus-merge algorithm, added even more complexity to the problem.

Consequently, we decided to simplify the commit history into a linear ordering by removing the least essential branches and merges. The rationale behind this decision is that each merge point captures changes that have happened on both branches; if the time difference from the branch to the subsequent merge is not too large, then the represented lifetime of the affected lines does not change substantially. (See analysis in the "Threats to Validity" Section.) An additional advantage of this approach is that it presents a commit sequence that is both topologically and temporarily ordered.

To obtain this linear ordering, we took the topological ordering of the project's commit graph and obtained the longest path in it. For directed acyclic graphs this path can be calculated in linear time, by having each vertex record its length as the maximum length of its parent neighbors plus one. Then the longest path can be obtained by traversing the graph along the vertices with the maximum recorded values. Figure 1 illustrates an example of a commit graph and its longest path. The simplification of history resulted in the reduction of examined commits from 3.106 million to 2.256 million, meaning that we processed about 73% of all commits.

## Lifetime tracking

One could in theory obtain an indication regarding the lifetime of individual lines by sampling the output of the Git's *blame* command at various time points. However, this process is computationally intensive and will only provide an approximation. To address these issues we designed an online algorithm and a corresponding open source tool

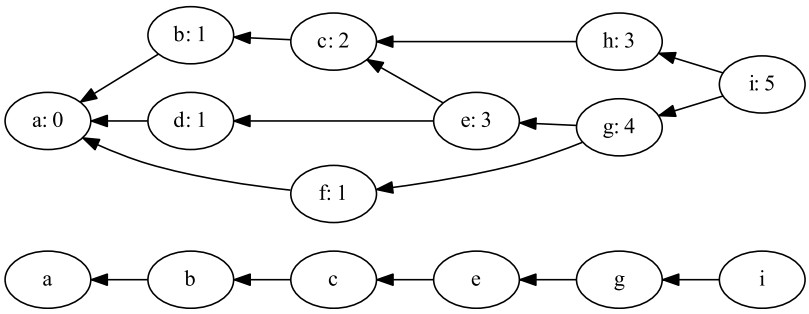

**Figure 1  Branch graph path length attributes and the longest path.**

(named *lifetime*) that continuously tracks the lifetime of code lines across successive commits.

Tracking the lifetime of individual code lines across code commits is not trivial. An earlier study that demonstrated the estimation of code decay in the Unix operating system source code over the period 1975–2015, employed the *git blame* command to record the age of lines at the point of each one of 71 releases (*Spinellis, 2016b*). Changes over the sampled releases in the cardinality of sets representing lines born at a specific point of time were then used to estimate the lifetime of the corresponding lines. However, this method is quite inaccurate, since the lifetime estimates are bound between the dates of two successive releases. Furthermore, it is also computationally expensive. The specific task required (on a massively parallel supercomputer) 9.9 core years CPU time, 3,815 cores, 7.6 TB RAM, and 588 GB of disk space. In fact, our case of 711 k files × 3.5 M commits would require two orders of magnitude more resources.

In common with most version control systems, Git can output the differences in a file between two commits as a series of line additions and removals. (Changes are represented as an addition and removal.) By default, this operation uses the popular Myers algorithm (*Myers, 1986*) to minimize the presented differences. In common with the work by *Zimmermann et al. (2006)*, we processed the output of Unix (Git) *diff*, rather than alternatives such as *LHDiff* (*Asaduzzaman et al., 2013a*), because *diff* operates fast and its output is machine-readable.

A line may appear in the output of a commit's differences for several reasons: (1) actual deletion—within a retained file or through a file's deletion, (2) changes in identifier names, (3) other non-whitespace changes, (4) movement to another part of the same file, (5) movement to another file, (6) change of indentation, or (7) other cosmetic—whitespace—changes. Reasons 1 and 3 are definitely signs of the line's death that are relevant to this study: a (most-probably) functional modification. Our methods also consider as a line death reasons 2 and 4, because it is difficult to identify such changes with the tools we employed. We deal with these potential shortcomings by expanding our methods to track changes of individual tokens and by measuring the effect of line moves. We were also able to continue tracking through their lifetime lines that change due to reasons 5–7. First, we set *git diff* to ignore all changes in a line's whitespace. This filtered out noise introduced by indentation adjustments induced through other changes as well as

**Listing 1 Example of git diff output.**

```
1 commit dfdcb9a67686[…]de95524b845d 1470512904
2
3 diff −−git a/main.c b/main.c
4 index 63161da..7a6f21d 100644
5 −−− a/main.c
6 +++ b/main.c
7 @@ −2,0 +3,2 @@
8 +#include "message.h"
9 +
10 @@ −5 +7 @@ main(int argc, char *argv[])
11 −        printf ("hello, world");
12 +        printf (MESSAGE);
13 diff −−git a/message.h b/message.h
14 new file mode 100644
15 index 0000000..be10a6e
16 −−−/dev/null
17 +++ b/message.h
18 @@ −0,0 +1 @@
19 +#define MESSAGE "hello, world"
```

changes in a file's tab (hard vs soft tabs) or end-of-line (the use of line feed and carriage return) representation. Second, we configured *git diff* to detect file renaming and copying, in order to follow cross-file code movements.

An example of the *git diff* output format processed by the *lifetime* tool we built appears in Listing 1. Line 1 is a custom commit header we employed, containing the commit's SHA identifier and its timestamp. The commit involves changes to two files; lines 3 and 13 show the old and new names of the files being compared. When a file is removed or newly added the new file name (in the case of removals) or old file name (for additions) is `/dev/null` (the Unix empty file). Lines 4–6 and 15–17 contain metadata that is not important for the task at hand. Lines 7–9 show the addition of two lines at the new file range +3,2 listed in the `@@` line. Lines 18–19 do the same for the second (newly-created) file. Lines 10–12 show a line's change: line 5 of the old version's file is replaced by line 7 in the new version's file. in the new file version. A series of extended headers can appear after the `diff` line to indicate file deletions, renames, and copies, which Git detects by applying heuristics in the compressed repository state snapshot stored for each commit.

The *lifetime* program[2] works by processing a series of *git diff* outputs (such as those detailed in the preceding paragraph) using the state machine depicted in Fig. 2. State transitions take place when input lines match the regular expressions shown in the diagram.

[2] Available in this study's source code package at https://doi.org/10.5281/zenodo.4319993 and also on GitHub https://github.com/dspinellis/code-lifetime.

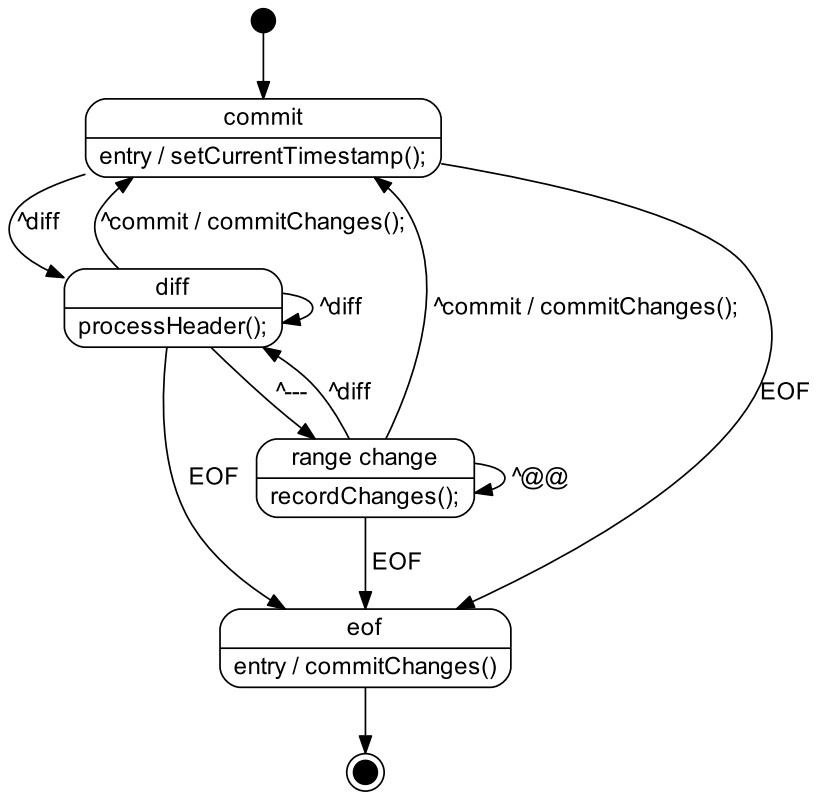

**Figure 2 State machine for *git diff* output processing.**

The operations can be expressed through the following notation.

- The timestamp $T_c$ associated with commit currently being processed.
- A partial function $T : (F, L) \rightarrow T_B$ mapping each integer-numbered line $L$ of file $F$ onto its birth time $T_B$.
- Another partial function $B : F \rightarrow v$, that yields true when a file $F$ contains binary data (e.g., an image).
- The last line of each file $F_E = \max(\{l : T(F, l) \neq \perp\})$.

The rules applied when processing the *git diff* data are the following.

1. For each code line numbered $L$ *added* to file $F$ that the program encounters (e.g., lines 8, 9, 12, 19 in Listing 1), it remaps existing timestamps from $L$ until the end of the file $E$ in the map $T$ it maintains to make space for the new line, and it inserts an entry in the timestamp map with the current timestamp $T_c$ (1470512904 in line 1 of Listing 1).

$$\forall l \in (L..F_E), T'(F, l + 1) = T(F, l)$$
$$T'(F, L) = T_c$$

2. For each code line numbered $L$ deleted from file $F$ that the program encounters (e.g., line 11 in Listing 1), it outputs a tuple with the line's birth time and the time of its demise,

$$(T(F, L), T_c)$$

and it remaps the timestamps of the lines from $L$ until the end of the file $E$ to close-up the gap of the deleted line.

$$\forall l \in (L..F_E - 1), T'(F, l) = T(F, l + 1)$$
$$T'(F, F_E) = \bot$$

3. When changes to a binary file $F$ are encountered, this file is marked as binary in a map $B$, and no further output is ever performed on operations on that file. This is needed because changes to binary files are not identified in terms of lines, so further changes when a file reverts to text format will not have correct timestamps to refer to.

$$B(F) = \text{true}$$

4. When a file $F_a$ is identified as copied to file $F_b$, new map entries are established with the original line birth dates and the binary file status is also transferred to the new file.

$$\forall l \in (1.. \max(F_{aE}, F_{bE})), T(F_b, l) = T(F_a, l)$$
$$B(F_b) = B(F_a)$$

5. When a file $F_a$ is identified as renamed to file $F_b$, new map entries are created as above, and the existing ones are removed.

6. After processing all commits, *lifetime* outputs tuples with the birth timestamps of all lines that are still alive and the word `alive`.

$$\{(T(F, L), \text{alive}) : T(F, L) \neq \bot\}$$

The processing is complicated by the fact that all change references to the existing state of a commit, refer to the state before any change has been applied to any of the files; changes are not supposed to be applied while processing a commit. For example, if a commit renames file $F_a$ to $F_b$ and $F_b$ to $F_a$ the names of the two files will be correctly swapped. Also, changes to a file that has been copied or renamed in the same commit refer to the name of the file before the corresponding operation. This complication is addressed by recording all changes in a queue as instructions to add or remove elements from the timestamp map $T$. When all elements of a commit have been processed, a routine replays the recorded changes on the current state to generate the new one.

Considerable effort was invested in making the *lifetime* program easy to test and troubleshoot. This was needed for three reasons. First, the output of *git-diff* seems to be only informally defined and involves many special cases. Second, tracking line timestamps by hand to verify the program's operation is a complex and error-prone process. Third, errors were encountered in the middle of processing data hundreds of gigabytes in size; isolating these errors proved to be a challenging task.

In order to help testing and troubleshooting the *lifetime* program supports eight command-line options that configure it to output diverse data regarding the processing it performs. A separate option can be used to terminate the processing at a specific commit, thus allowing the examination of the data at the given time point. The most important of

the debug options, modifies the program's operation to store in the map $T$ the complete content of each added line, rather than the current timestamp $T_c$. Then, when a line is removed, it is compared with the map's content to verify that the two match. Any differences signify a failure in the process to record the changes. Such differences allowed us to find that the output of `git log` and `git format-patch` were not trustworthy enough for our purposes. Furthermore, the same debug option uses the map's contents to reconstruct a copy of the project's file tree, when all its commits have been processed. Comparing the file tree against a checked-out version of the project allows the end-to-end verification of the program's operation. The *lifetime* program is accompanied by two Git repositories containing tens of diverse commit cases. A test script is used to compare the reconstructed state against the original one at the end of each commit.

## Analysis of individual tokens

Although lines of code are often used to measure software and its evolution, tracking changes at the level of lines can threaten the results' validity. Specifically, small changes, such as renaming an identifier, will appear to change many lines. In addition, a line may appear to change through edits unrelated to it, such as the addition of a brace when a statement is added below it. Consequently, it would be valuable to track evolution at the level of individual tokens rather than lines.

We designed and implemented a process and tools to track the birth and demise of individual tokens based on an idea by *German, Adams & Stewart (2019)*. This involves creating a synthetic Git repository where files are stored as one token per line. The setup can be traced to the more general concept of using a synthetic Git repository to track arbitrary fine-grained elements (*Hata, Mizuno & Kikuno, 2011*). The downloaded repositories amount to 30 GB and the synthetic ones to 32 GB. Tracking changes between revisions in such a repository will show the addition and removal of individual tokens, rather than complete lines. All other workflow and tools can remain the same.

We created tokenized versions of the selected repositories through two tools: a file tokenizer and a repository tokenizer. The repository tokenizer is a Perl script that acts as a filter between a *git fast-export* and a *git fast-import* command. It reads the dump of the original repository generated by the *git fast-export* command, queueing file content blobs it encounters, while passing the remaining data unchanged to its output. When it reads a commit packet, it matches the file extensions of the committed files against previously encountered blobs. For any blob whose file extension matches the languages supported by the file tokenizer, the repository tokenizer invokes the file tokenizer to convert the file into tokens, dumping the tokenized results on its output as the corresponding blob.

To tokenize the contents of each file, we used the *tokenizer* tool, which splits its input into tokens using simple look-ahead lexical analysis (*Spinellis, 2019*). Support for each language is provided through a separate lexical analyzer to cater for differences in operators, reserved words, commenting, and string delimiters. Through command-line options we directed the file tokenizer to split its input into a token per line replacing the

content of strings and comments with an ellipsis (…), thus also allowing us to ignore non-code changes.

## Analysis of moved lines

Given that the file differencing program we employed will not report line moves within the same file, we attempted to quantify the effect of this behavior on our results. For this, we developed a tool that uses Heckel's linear time differencing algorithm (*Heckel, 1978*), which does attempt to locate line moves. Although the output of this program is not suitable for running the fully fledged analysis, its summary of added and removed lines can be tallied against that of the *git diff* program to compare their performance in detecting lines that have not changed. By configuring *git diff* to run the alternative program between all successive revisions, we found that Heckel's algorithm, despite taking into account line moves, reports 2.2% more line additions and deletions than Git's stock algorithm. While differencing algorithms can always be tweaked to handle elaborate special cases, this result indicates that the differencing program we employed for the study works pretty hard to identify a competitively small set of differences, and that taking into account line moves using Heckel's algorithm would reduce the accuracy of our results by failing to track about 2% more lines.

## Effect of the histogram algorithm

Following the recommendation of a recent systematic survey of studies that use *diff* (*Nugroho, Hata & Matsumoto, 2020*), we also examined whether the use of Git's Histogram difference algorithm would substantially alter our results. Using a process similar to that described in the "Analysis of Moved Lines", we measured the differences in the reported added and deleted lines between the Myers and the Histogram algorithms. Both differences were below 0.5%: 0.28% for deletions and −0.45% for additions. The effect's small size is not surprising, because the Histogram algorithm mainly improves the readability of the provided patch.

## Statistical analysis

If we had the time of birth and the time of death for each line of code and token in a project we could estimate the median lifespan of the line or token directly, by calculating lifespans and finding their median value. We are interested in the *median* and not the *mean*, because lifetimes may be skewed, so the mean, or average, would not give a representative metric.

Unfortunately, we do not have the lifespans of all lines and tokens that we have tracked in the repositories we have examined. We do have their birth timestamps, but there are many lines and tokens that are still alive at the end of our follow-up period: these are the lines and the tokens that are part of the code base of a project at the last time we check. Their lifespans are *right censored*; they extend to the future. Across projects, the mean of the percentage of right censored lines is 29.97% and the median is 28.25%; for tokens, the corresponding values are 33.38% and 30.87% respectively.

To estimate the median lifespan under such circumstances we use the Kaplan–Meier or product-limit survival estimate (*Kaplan & Meier, 1958*). If our measurements take place at times $t_1 < t_2 < \ldots < t_n$ and at time $t_i$ we have $n_i$ that are alive, of which $d_i$ die right at that moment, then the probability of being alive at time $t_i$ is given by:

$$S(t_i) = S(t_{i-1})\left(\frac{n_i - d_i}{n_i}\right)$$

The recursive definition assumes $t_0 < t_1$ and $S(t_0) = 1$.

In our data, lines and tokens are born at different times during a project. Since we are interested in their lifespans and not at the chronological times of birth and death, we work only with the differences between birth and death timestamps. That means that the times $t_i$ are time offsets; time 0 is the birth time for all lines. For example, if we have a line with a lifespan from $t_i$ to $t_j$ we take for its death time the difference $t_j - t_i$. For the lines and the tokens that are right censored, we assign as their death timestamp the latest timestamp in the project. We also flag them as being alive, which means that in the Kaplan–Meier estimation their lifespan will be taken to be at least until the latest project timestamp. Note that we do not have censoring due to other causes, for example, a line being "lost" somewhere in the project's timeline, without being able to follow it up.

The function $S(t)$ is stepwise, with constant values between the different $t_i$. It estimates the *survival function* of a data set, which is formally defined as follows: $S(t)$ is the probability that an individual (line of code in our case) survives longer than a time $t$:

$$S(t) = P(T > t)$$

In the above, $T$ is a variable indicating the time of death. We would also like to know what is the risk of dying at $t$. For this, we have to turn to the *hazard function*, or *hazard rate*, $h(t)$, which is the rate at which an individual that has made it to time $t$ will die within an interval $\delta t$ that tends to zero:

$$h(t) = \lim_{\Delta t \to 0} \frac{P(t \le T < \Delta t | T \ge t)}{\Delta t}$$

We have three alternative hypotheses regarding the hazard function:

- Individuals run the same, constant, risk of death at each time $t$.
- Individuals run a higher risk of death when they are young; these are populations whose demographics are characterized by high infant mortality.
- Individuals run a higher risk of death when they are old; these are populations whose frailty increases with age (senescence).

To test these hypotheses we will check whether the hazard function for lines of code follows a Weibull distribution, a standard parametric model in survival analysis that has

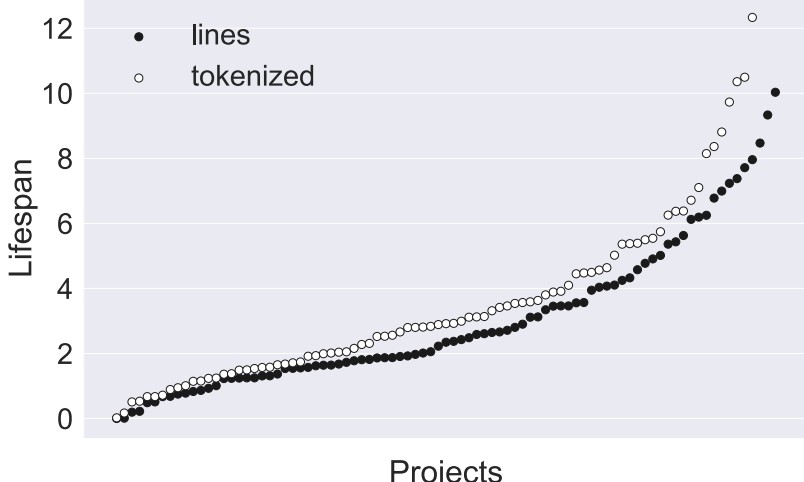

**Figure 3 Kaplan–Meier median lifespan estimates.** Lifespan estimates per project, in increasing order, calculated for lines and for individual tokens.

been used widely in science and engineering (*Padgett, 2011*). The Weibull distribution specifies the following hazard rate, with two parameters $\lambda > 0$, $\alpha > 0$

$$h(t; \lambda, \alpha) = \alpha \lambda t^{\alpha-1}$$

The corresponding survival function is:

$$S(t; \lambda, \alpha) = e^{-\lambda t^{\alpha}}$$

The parameter $\alpha$ is called the *shape parameter* and the parameter $\lambda$ is called the *scale parameter*. Together they determine the form of the corresponding Weibull probability density function $f(t; \lambda, \alpha) = \alpha \lambda t^{\alpha-1} e^{-\lambda t^{\alpha}}$. The parameter $\lambda$ stretches or contracts the distribution along the $x$ axis. There are three different cases for the parameter $\alpha$:

- If $\alpha < 1$, the hazard rate decreases over time.
- If $\alpha = 1$, the hazard rate is constant.
- If $\alpha > 1$, the hazard rate increases with time.

The three alternatives for $\alpha$ mirror the three hypotheses we want to check and can be the basis of the statistical analysis of the code aging process.

## RESULTS AND DISCUSSION

### RQ1

The Kaplan–Meier estimate provided the median lifespan for the projects we examined. This is the point in time at which 50% of the population has died. Figure 3 shows the Kaplan–Meier survival functions for all projects, in increasing order.

The minimum line median lifespan, at 0.000105 years (about 0.9 h), is for the project *HandBrake*; the maximum line median lifespan, at 10.03 years, is for *collectd*, while for *torque* and *boto* the lifespan could not be calculated, because not enough lines had died by

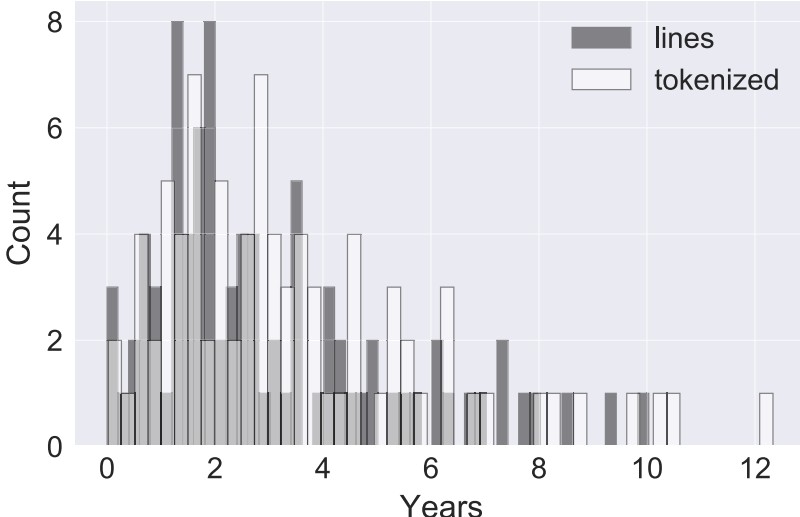

**Figure 4 Histogram of median lifespans.** Distributions of median lifespans of lines and individual tokens.                               

the end of the data collection period to be able to get to the 50% point. We investigated the extremely low value for *HandBrake*. It appears that the project features large commits and incorporates the entire Mac Sparkle framework within the repository.

Turning to tokens, the minimum median lifespan was for *odoo* at 0.02 years. There were four projects for which no median lifespan could be calculated: *thrift*, *mpc-hc*, *boto*, *collectd*, *torque*. The maximum median lifespan that could be calculated was for *docuwiki*, at 12.33 years.

Taking all line results together, the median of the median lifespans is at 2.37 years, while the 25% percentile is at 1.54 years and the 75% percentile is at 4.25 years. For tokens, the corresponding median is 2.93 years, the 25% percentile 1.67 years, and the 75% percentile 5.36 years. These results indicate that lines and their individual tokens are durable rather than perishable, with lifespans measured in years rather than days. Figure 4 shows the histograms of the median lifespans.

The growth of projects is punctuated with bursts of additions and deletes; these occur when a large body of code is imported or removed *en masse* from the project. We examined whether the estimates would change if we remove outliers. We therefore carried out the same statistics after removing the lines and tokens that where introduced in commits that were in the top 1% of commit size in every project. The line median lifespan moved to 2.54 years, an increase of 7.17%; the token median lifespan moved to 3.18 years, an increase of 8.53%. That is not trivial, but it does not change the overall picture.

To determine whether the differences in the medians between the line-oriented and the tokenized data can be explained away by chance, we carried out a Wilcoxon signed-rank test. The null hypothesis was that the two median populations come from the same distribution. The test allowed us to reject the null hypothesis with high probability ($p$-value close to zero). It follows that code tokens lead longer lives than code lines;

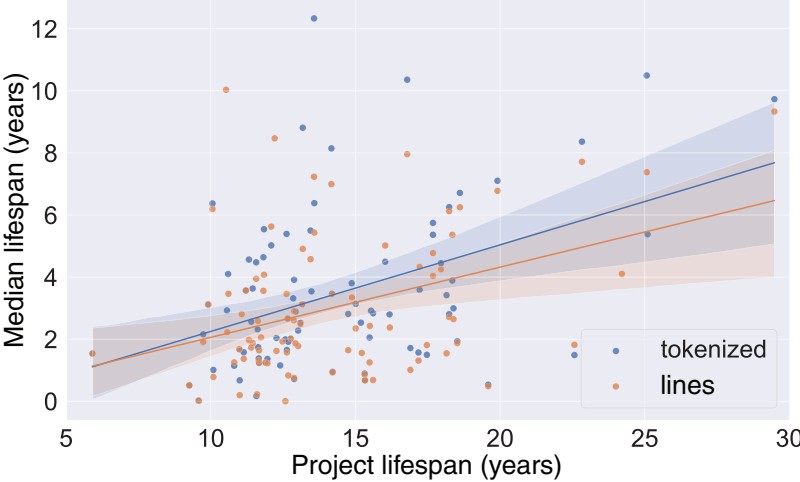

**Figure 5 Line median lifespan on project lifespan.** Scatterplots and regression lines of the lifespan of each project vs the median lifespan.

after all, every token that changes affects the line in which it belongs, but the opposite does not hold.

As project lifespans vary, the variability of code lifespans may be explained by the variability of project lifespans: code in longer-lived projects may live longer than code in younger projects. To investigate that, we performed correlation tests between median line lifespans and project lifespans. The Spearman correlation test for lines produced $\rho = 0.29$ ($p < 0.01$), which indicates a slight monotonically increasing relationship between median life lifespan and project lifespan. The Pearson correlation test produced $r = 0.39$ ($p \ll 0.01$); the difference with the Spearman result can be explained if the relationship is monotonically increasing, but not linear. For tokenized data, the correlation was a bit stronger, with Spearman $\rho = 0.37$ ($p \ll 0.01$) and Pearson $r = 0.44$ ($p \ll 0.01$). Figure 5 shows a scatterplot of line and tokens median lifespans with a regression line; the regression coefficient for lines is 0.23 and for tokens is 0.28. In all, although the effect of project age is statistically significant, its effect on the longevity of code is small.

## RQ2

Moving beyond the estimates of median line lifespan, we checked the three hypotheses on hazard rates by fitting a Weibull distribution to each project's data. The fit was performed on the full line data of each project; we are interested in the fitted Weibull $\alpha$ parameter that controls the shape of the distribution and therefore the evolution of the hazard rate. The results of the fit showed that for all projects the $\alpha$ parameter is less than one, indicating a process with high infant mortality. Figure 6 shows the Weibull fitted distributions for all projects, each line being a project. The median of $\alpha$ is 0.52, while the 25% percentile is 0.43 and the 75% percentile 0.63. The situation is almost the same if we do the same analysis for tokens. Two projects have $\alpha \geq 1$ (by a whisker, *ojs* with $\alpha = 1.01$ and *xmbc* with $\alpha = 1.07$). The median is 0.53, the 25% percentile 0.43 and the 75% percentile 0.70.

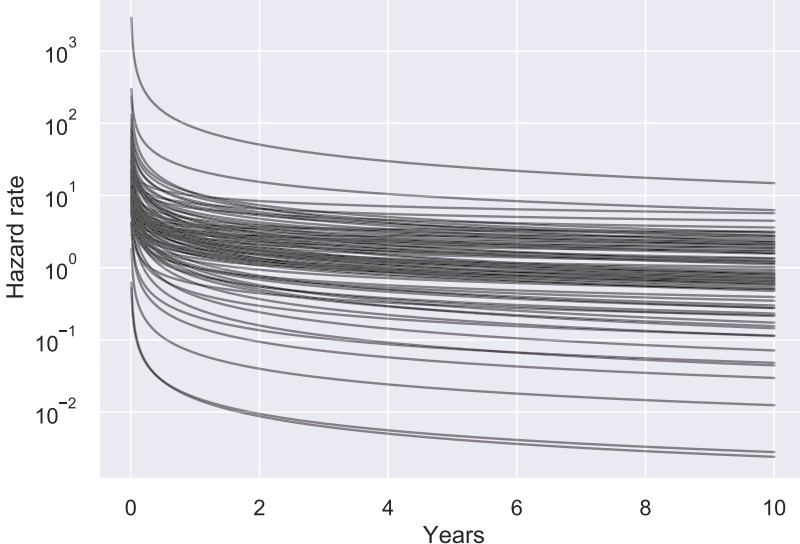

**Figure 6 Hazard rates of lines of code per project.** For all projects, the hazard rate for lines decreases with time, i.e., older lines run a smaller risk of dying.

From the above, it follows that young lines run higher risks. Old lines die as well, but younger ones at higher rates. This suggests a software development process where lines that are introduced into the code base of the project are subject to more change pressures. A line that has just been committed may not have been as thoroughly tested as older lines; it may need to be modified to accommodate factors that had not been foreseen; lines just added may impact more those recently introduced than parts of the older code base.

Conversely, old lines seem to have proved their mettle. They have survived a long time and they are less likely to suffer changes than young lines. In a more negative light, old lines may gain a "don't touch" status, where developers are wary to change anything that works, which therefore lives on.

Whichever may hold, that a line lives on because it is really valuable or because nobody dares to change it, developers should be aware that they work for the long term. A line of code may live for years, well beyond the developers' involvement with a project or their ability to remember the rationale behind a cryptic choice. Consequently, our findings provide one more reason for writing clear and well-documented code.

Our findings also support the need to manage and perform what have been called *anti-regressive* changes (*Lehman, 1978*) to the software (effort required to combat the growth in complexity) in order to avoid the accumulation of technical debt (*Kruchten, Nord & Ozkaya, 2012*). Code lines that live long are likely to become out of sync with respect to the software's evolving architecture, design, APIs, third-party libraries, language standards, as well as coding conventions and practices. As we have shown, such lines are not very likely to go away. Consequently, it is required to find those code lines that need care and bring them up to scratch. This is typically accomplished through the detection of code smells and the corresponding refactoring of code (*Fowler, 2000*).

### RQ3a

We investigated whether particular features of lines are conducive to more changes. We ran a linear multiple regression model for each project with the lifespan of a line as the dependent variable and as independent variables the length of the line, the indentation, the number of strings in the line, whether it (or part of it) is a comment, the number of commas, the number of brackets, the number of access operators (method and pointer), the number of assignments, the number of scope delimiters, the number of array accesses, and the number of logical operators. The elements we tested point to code smells or other code features that may make the code less stable, affecting its lifetime. We selected the aforementioned features for the following reasons. A large number of brackets may indicate complicated conditions or expressions, or long statements, which are a known code smell (*Sharma, Fragkoulis & Spinellis, 2016*). A large number of commas may indicate a long parameter list smell (*Mäntylä & Lassenius, 2006*). Strings may indicate the entanglement of presentation elements with business logic (*Nguyen et al., 2012*).

The results showed a very low fit ($R^2 < 0.1$) for all projects, apart from *canu* with $R^2 = 0.44$, *HandBrake*, with $R^2 = 0.29$, and *pyparallel*, with $R^2 = 0.11$. The regression coefficients were found with very small $p$ values, which indicates that the influence they have on the lifespan cannot be explained away by chance, but the whole linear model, and therefore each particular predictor in it, accounts for a tiny part of lifetime variance.

### RQ3b

To conduct a similar analysis for tokens, we divided tokens in four types: identifiers (391 million), numbers (91 million), keywords (112 million), and other tokens (mainly operators and punctuation—723 million). We ran pairwise Mann–Whitney $U$ tests between the lifetimes of different token types for each project. The distributions of the lifetimes of token types per project are different in the vast majority of projects (ranging from the distributions of 86/89 projects for identifiers vs other tokens to the distributions of 82/89 projects for keywords vs numbers). However, when we take the medians of the lifetimes of different token types for all projects, their distributions are then all indistinguishable. As for lines, we could not determine that some particular types of tokens are associated with longer lifetimes across projects.

### RQ3c

A different factor that may influence the lifespan of a line is the committer who enters, alters, or deletes a line. We examined possible correlations between the lifespan and the number of developer commits in the project and between the experience and the tenure of the developer in the project. For this, we looked at commits in the middle year of the examined period (2012), thus providing at least 4.5 years time to gather line and developer data and then another 4.5 for lines to disappear. We used the number of a developer's project commits until each examined commit as a measure of a developer's experience, and the difference in time between the developer's first project commit and the examined one as a measure of the developer's tenure in the project. We carried out both Spearman and Pearson correlation tests to examine the relationship between line lifetime and the

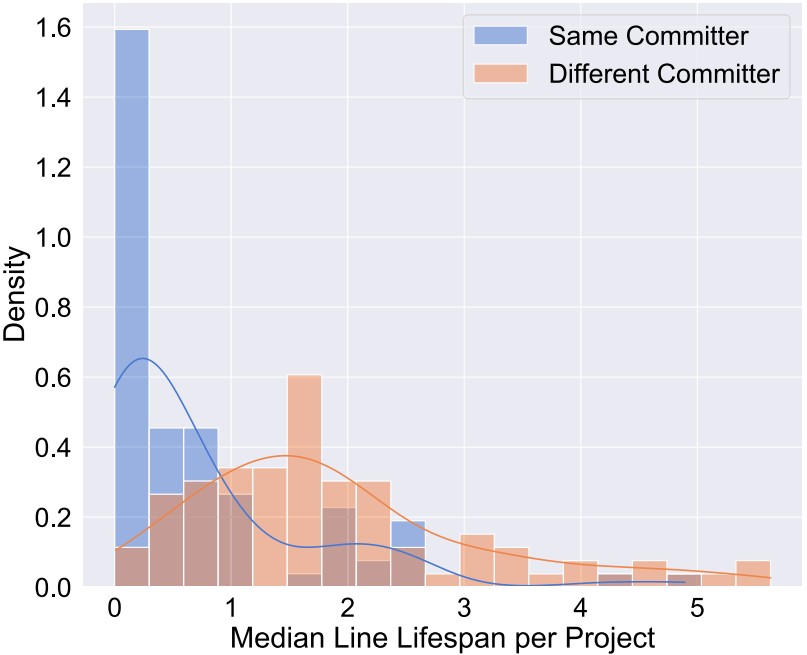

**Figure 7 Per project lifetime medians for same and different committers.**

experience and tenure of the developer who added the line. We could not identify a single rule across projects. In some projects, committer activity and tenure appear to be positive correlated with line lifespan, in other projects they appear to be negative correlated, and in most projects the correlation seems to be weak: the median is close to zero. The situation changes when we examine the lifetime of lines vs the experience and tenure of the author who removed them: we find that the lifetime is positively correlated with developer experience, that is, more experienced developers remove longer-lived lines. The median of the correlation of line lifetime and developer experience across projects is 0.27 (Spearman) and 0.24 (Pearson) for $p < 0.05$; for the correlation of line lifetime and developer tenure the medians are 0.42 (Spearman) and 0.33 (Pearson) for $p < 0.05$.

Alternatively, the above can mean that it takes experienced developers to remove a long-lived line, bringing us back to the "don't touch" status. The "don't touch" status also hints at a different facet of the way lines are handled. Could it be that lines are more likely to be changed or deleted by the same developer who entered them into a project in the first place, rather than by a different committer? We contrasted, for each project, the lifetimes of lines that are changed by the same developer against those that are changed by a different one. The two distributions are different (checking with the Mann–Whitney $U$ test) for all projects except *drush* and *grails-core*. In most projects, the median lifetime of lines removed by the same author who entered them is less than the median lifetime of lines removed by a different author (81/89); and similarly for the means (85/89). In short, lines are more likely to be touched by their original author (see also Fig. 7).

**Table 2  Languages and Kaplan–Meier (KM) estimates.**

| Language | # Projects | Median Line KM | Median Token KM |
|---|---|---|---|
| PHP | 16 | 1.93 | 2.42 |
| C++ | 17 | 1.86 | 2.81 |
| Python | 12 | 2.75 | 3.61 |
| Java | 18 | 2.45 | 2.74 |
| C | 25 | 2.80 | 3.63 |

## RQ3d

We investigated whether project size affects the longevity of lines and tokens. We checked the number of lines and the number of tokens for all commits, using both Pearson and Spearman correlations. We found only slight positive correlations, $r = 0.35$ ($p < 0.01$), for the Pearson correlation (but not the Spearman) for both the number of lines and the number of tokens in a project. Of course, the size of the project may be related to its age and indeed the results are concordant with our preceding investigation on code lifespans and project lifespans.

## RQ3e

Turning to programing languages, although we expected to find greater lifespans in languages with features that promote modularity, we did not detect that. Table 2 shows that median lifetime estimates over projects grouped by programing language (excluding a single project in C#). If anything, we see that, for instance, C exhibits greater lifespans than C++. However, note that none of the differences between programing languages was statistically significant at the 0.01 level using the Mann–Whitney $U$ test.

# THREATS TO VALIDITY

## Internal validity

Thankfully, by basing our study on historical data, many threats that typically appear in evolving experiments, such as design contamination, experimental mortality, and maturation, can be ruled out. The main remaining threats are associated with confounding factors, noise in the data, commit granularity, file differencing, and statistical methods.

An important consideration is that the independent variable we used in our study, a code line's age, can encompass many other variables. Specifically with the passage of time, the *number of faults* in a line will decrease as these are winnowed out (*Ozment & Schechter, 2006*), the developers' familiarity will increase as they read it again and again, and the line's afferent couplings may increase as other code may depend on its elements.

Another factor is noise in the data we used. Although we were careful to include (through simple measures) in our study what has been termed *engineered software projects* (*Munaiah et al., 2017*), we cannot exclude the possibility that the underlying commit data contain infelicities that may influence our results. These include the addition or removal of large third-party software systems, wrongly set commit dates, history rewrites

performed after the code was written, and errors introduced when one type of repository (e.g., CVS) gets converted into Git.

Third-party code changes can have a significant effect on software evolution. *Gall et al. (1997)* conducted an empirical study on a large telecommunication switching system, and identified important differences in the patterns of software evolution over time of the whole system vs its subsystems. A similar strategy of separately examining the growth of subsystems of large software projects has been followed by *González-Barahona et al. (2014)* and by *González-Barahona et al. (2009)*. Other studies of software evolution have also identified the ripples caused by the inclusion or removal of third-party components (*Robles, González-Barahona & Herraiz, 2005*), and some, such as the one by *Hatton, Spinellis & Van Genuchten (2017)*, have attempted to address the issue by filtering them out. As we have not performed such filtering, these changes may affect our reported results. On the other hand, filtering introduces another threat to validity due to the subjective nature of the required decisions or parameters.

A related factor is the granularity of the studied commits. Our study is missing many intermediary commits, first because we removed about 32% through history simplification, and second because many others may have occurred in third-party repositories and then pushed upstream as a single commit (*German, Adams & Hassan, 2016*). One could argue that the effects of history simplification should cancel out: lines would on average appear later and also disappear later. Nevertheless, to quantify the effect of history simplification, we measured the interval between commits in both the complete tree and the simplified longest path. As expected, the longer time paths upstream from merges in the complete tree, which were simplified away in the linear path, gave the tree a longer interval between commits (a median of 40 min) than its longest path (27 min). However, the difference between the median value of the two intervals (13 min) is five orders of magnitude smaller than the line lifespan we report, making any effect negligible.

The use of Git to list the differences between two file versions is also a threat. First, the employed file difference algorithm (*Myers, 1986*) will display a movement of code as a deletion and an insertion. Then, relatively minor changes, such as the renaming of an identifier, will appear as line deletions and insertions, which may skew the results toward higher infant mortality. We examined the effect of these two issues through methods described in the sections on the analysis of individual tokens and moved lines. Also, the detection of file renaming and copying is based on a heuristic and a threshold. We used the default thresholds, only increasing the number of files that would be checked for copies; there may conceivably be better values to use.

A related issue is that our investigation focuses on individual code lines. We do not take context into account. A line that is moved from one place to the next impacts both places and may cause cascading changes in the lifespans of other lines. However, the same applies in traditional survival analysis: deaths may be related to other deaths (e.g., via disease); we are interested in the lifespan of lines, no matter the relationships that may exist between them.

Given that we used custom-developed tools to track the birth and death of lines of code, human error is an inevitable factor. We tried to minimize this through numerous test cases, manual verification, and the use of internal consistency checks.

Turning to our statistical methods, we have used two statistical techniques to answer two different, but related questions. We used the Kaplan–Meier estimator to investigate the median lifespan of code in projects, and a Weibull process to investigate the overall aging process. The Kaplan–Meier estimator provided us with approximations of the survival functions, while the Weibull fit gave us approximations of the hazard functions. We assumed that the hazard rate is characterized by a Weibull function, because the Weibull distribution is a popular model for several related process such as component failure rates and Weibull covers different aging processes depending on the value of $\alpha$. Moreover, we are not interested in the exact values of the parameters of the Weibull distribution, but in the relation of $\alpha$ to 1, where we found consistent results. The remarkable agreement in the shape of the Weibull distributions among many diverse projects (Fig. 6) leads us to believe that our findings are reproducible and generalizable. We examined whether the lognormal distribution, which is also often used in failure models, would be a better fit for our data. To compare the two models, Weibull and lognormal, we used the Akaike Information Criterion (AIC), defined as $\text{AIC} = 2k - 2\log(\hat{L})$, where $k$ is the number of parameters of the model and $\hat{L}$ is the maximum likelihood of the model. A lower AIC value corresponds to a better fit, as this maximizes the goodness of fit, given by the log-likelihood, but penalizes the complexity of the model, given by the number of parameters. We found overwhelmingly that the Weibull distribution was a better fit. Only five projects had better fit with the lognormal distribution when we examined the lines, and six projects had better fit with the lognormal distribution when we examined the tokens (four projects were the same). We used the Python lifelines package for calculating the estimates and comparing the distributions (*Davidson-Pilon et al., 2020*).

## External validity

The generalizability of our findings is threatened by our choice of analyzed projects. Although we included projects from diverse development communities, written in numerous programing languages, and serving many different application areas, we cannot claim that our choice represents adequately all software development. In particular, we feel that our sample excludes or underrepresents the following software types: small software projects, projects developed with tightly managed or formal processes, proprietary and bespoke systems, projects written in programing languages not favored by the open source community, and systems that target specific application domains rather than the provision of systems infrastructure.

More importantly, our findings are based on large, successful projects that have run for several years. There are many more projects that are discontinued after a short period of time, for any reason. All lines of code in these projects freeze at an early stage of what could have been a longer period of evolution. Therefore our findings cannot be generalized to all software development—this would be an instance of *survival bias*,

reaching conclusions for all the population based only on the characteristics of the survivors. That said, people usually aspire to create successful, long-lasting projects, so our findings are pertinent for those projects that want to achieve longevity.

## RELATED WORK

All living beings degenerate and die with age. The origin of senescence, however, remains an unsolved problem for biologists (*Kirkwood & Austad, 2000*). Likewise, many software components evolve, age, and are eventually removed or replaced. This section presents related work regarding the fields of software evolution, aging, and decay, and records empirical studies that use the statistical method of survival analysis (*Elandt-Johnson & Johnson, 1990*; *Klein & Moeschberger, 2003*).

The process of **software evolution** refers to the modification and adaptation of software programs so that programs can survive as their environment changes. The software evolution laws of *Lehman (1980)* describe the constraints practitioners should take into account to continuously adapt actively used software systems. A detailed literature review regarding Lehman's software evolution laws has been conducted by *Herraiz et al. (2013)*. Many empirical studies focus on predictive models of software projects' evolution at macroscopic scale. Relevant studies have looked at long-term sustainability factors in the evolution of LibreOffice (*Gamalielsson & Lundell, 2014*), the change of program dependencies in the Apache ecosystem (*Bavota et al., 2013*), and the early identification of source code evolution pattern in open source projects (*Karus, 2013*). Additionally, many empirical studies examine software evolution at the microscopic level, considering the evolution of source-code elements such as methods. In particular, *Bevan et al. (2005)* developed *Kenyon*, which supports different types of stratigraphic software evolution research, ranging from code feature evolution to dependency graph-based maintenance. *Zimmermann (2006)* presented *APFEL* for fine-grained processing of source code elements such as tokens, method calls, exceptions, and variable usages. *Hata, Mizuno & Kikuno (2011)* introduced *Historage*, which provides entire histories of fine-grained entities in Java, such as methods, constructors, and fields. This tool has been applied to quantitatively evaluate the remaining change identification of open source software projects.

The term software aging was coined by *Parnas (1994)* and refers to the idea that programs, like people, are getting old. According to Parnas, software aging happens for two reasons: (1) software fails to adapt to changing needs, and (2) software changes but in an inappropriate way (addition of bad fixes and features). Given, however, that it is infeasible for developers to prevent software evolution and, consequently, software degradation, researchers attempt to limit program damages by predicting the software's lifetime and inventing rejuvenation approaches (*Karus & Dumas, 2012*; *Li et al., 2011*; *Qin et al., 2005*; *Robillard & Murphy, 2007*; *Salfner, Lenk & Malek, 2010*). In the field of software aging, empirical studies have been conducted on the identification of aging trends. In particular, *Robles et al. (2005)* found that a system becomes "old" when it turns five. The authors also defined the absolute 5-year aging index to compare the relative aging of different projects. Finally, *Cotroneo, Natella & Pietrantuono (2013)* developed an approach that predicts the location of aging-related bugs using software complexity

metrics and machine learning algorithms. They found that the most significant signs of software aging manifest themselves as: leaked memory, unterminated threads, unreleased files and locks, numerical errors, and disk fragmentation.

As software evolves, developers should overcome software erosion by fighting software decay. A significant research body is also devoted to this field. *Eick et al. (2001)* used the term code decay to describe the situation where evolving software increasingly hinders software maintenance. The authors, also, proposed measurements (code decay indices) as decay predictors. For their study, they statically analyzed millions of lines of a fifteen-year old telephone switching software system. Similarly to our work, the authors tracked added and deleted source code lines. However, they did not use survival analysis and they examined a single project to find particular code decay factors. Additionally, *Arisholm & Sjøberg (2000)* proposed a framework for the empirical assessment of changeability decay and *Araújo, Monteiro & Travassos (2012)* built a software decay model regarding software deterioration causes.

Extensive work has been done on identifying and tracking code changes. *Kim & Notkin (2006)* were the first that defined the problem of matching code elements between two program versions based on syntactic and textual similarity. To compute the difference between two programs several tools have been implemented. *Canfora, Cerulo & Penta (2007)* developed a technique that combines Space Vector Models and the *Levenshtein* edit distance for finding CVS/SVN differences that occur due to line additions or deletions, as well as due to line modifications. Furthermore, the *LHDiff* tool implements language-independent techniques to track how source code lines evolve across different versions of a software system (*Asaduzzaman et al., 2013b*). The tool uses the *simhash* technique together with heuristics. In addition, the *GumTree* tool identifies edits in scripts when moving code in version repositories (*Falleri et al., 2014*). This tool is based on a abstract syntax tree (AST) differencing algorithm. The *ChangeDistiller* is another differencing tool that is based on a tree differencing algorithm for fine-grained source code change extraction (*Fluri et al., 2007*). To represent how lines evolve over time in source code version repositories researchers have also used annotation graphs (*Zimmermann et al., 2006*). More recently, *Servant & Jones (2017)* proposed a fine-grained model based on optimizing the *Levenshtein* difference between lines of successive versions. Finally, *CVSscan* is a visual tool for representing software evolution based on the tracking of line-based source code changes extracted by using *Unix's diff* (*Voinea, Telea & Van Wijk, 2005*). The *lifetime* tool we present here, balances computational cost with accuracy by processing a series of *git diff* outputs and uses a state machine for the parsing of their output.

Other researchers have also employed the tools of survival analysis in software (*Elandt-Johnson & Johnson, 1990*; *Klein & Moeschberger, 2003*). *Sentas, Angelis & Stamelos (2008)* developed a statistical framework based on survival analysis and the Kaplan–Meier estimator (*Kaplan & Meier, 1958*) to predict the duration of software projects and the factors that affect it. The authors applied their approach on proprietary software projects taking into account industrial factors that have an impact on a project's lifetime. They found that the median duration of the examined projects is 14 months. Similarly,

*Samoladas, Angelis & Stamelos (2010)* applied survival analysis on 1,147 open source projects to forecast software evolution trends. The authors observed that projects that existed more than ten years ago continue to evolve. Comparably, our insights confirm that code in long-lived projects lives longer. *Scanniello (2011)* used the Kaplan–Meier estimator on five software projects (at method level) to investigate how dead code affects software evolution.

Our finding regarding the lower hazard of older lines is mirrored by *Zheng et al. (2008)* who report that in Gentoo Linux packages, a network graph new node is connected to an old node with a probability that depends not only on the degree but also on the age of the old node. Other survival analysis studies include the one by *Claes et al. (2015)* on the longevity of Debian packages with conflicts and the one by *Goeminne & Mens (2015)* on the survival and influence of five Java database frameworks. To the best of our knowledge, this article is the first work that uses survival analysis to track the birth and death of code lines and tokens over periods that span decades, and presents a theoretical and statistical model regarding the aging process of code lines.

Another research strand that the study of the evolution of fine-grained code elements is related to includes genetic improvement. Genetic improvement (GI) uses automated search (i.e., optimization and machine learning techniques) in order to improve existing software. Typically, GI involves making small changes or edits (also known as *mutations*) in source-code elements (i.e., lines of code or tokens) to improve existing software. Topics covered by GI research include program transformation, approximate computing, and program repair (*Petke et al., 2018*). As an example, *Petke et al. (2014)* apply GI with automated code transplantation, by mutating the code at the level of lines of source code, to improve software performance. Additionally, *Barr et al. (2014)* introduced the *plastic surgery hypothesis*, which states that changes to a codebase refer to source code elements that already exist in the codebase at the time of a change. Related work (*Nguyen et al., 2013*; *Goues, Forrest & Weimer, 2013*) considers repetitiveness of code changes (abstracted to abstract syntax trees) that is associated with the plastic surgery hypothesis. Furthermore, *Martinez, Weimer & Monperrus (2014)* consider changes that could be constructed based on existing code snippets. Therefore, the study of the evolution of code elements, such as code lines or tokens that we take into account here, could help, in the future, in the guidance of software improvement based on evolutionary approaches.

## CONCLUSIONS AND IMPLICATIONS

When we began working on this study, we did not know whether code lines were durable or perishable and whether their demise was a result of infant mortality or senescence. By devising a method and implementing tools to follow source code lines from 89 revision control repositories from their birth to their demise, we were able to arrive at the answers through the statistical analysis of 3.3 billion source code lifetime events. We found that code lines are durable with a median lifespan of about 2.4 years, the corresponding median for the tokens is 2.9 years, and that the associated hazard rate decreases over time following a Weibull distribution; that is, young lines are more likely to

be modified or deleted. We investigated whether line and token longevity are associated with particular line and token features, developer experience and tenure in a project, and programing language. Our results did not show strong patterns, indicating that line and token longevity may be the result of a complex interaction of various, potentially context specific, factors. Project age and project size had a small correlation with code longevity.

On the practical front, our model, suitably calibrated, can provide input for estimating maintenance effort, while the corresponding tool could aid the management of technical debt and the risk assessment of software vulnerabilities. Our model derives statistical estimates of lifespan estimates and hazard rates, based on the source code of projects. They can be run on other projects, apart from the ones we used, to give the calibrated figures for them. Knowing how lines and token age (or churn) in a project may help in managing technical debt and risk assessment (*Ozment & Schechter, 2006*; *Shin et al., 2010*). For example, a large number of long-lived lines/tokens can be a sign of stability or "don't touch" status. Moreover, our regression model of lines lifespan vs project lifespan can be used against a particular project to gauge where it stands, or whether (perhaps problematically) it is an outlier. All these potential uses need to be empirically validated in future studies.

On the research front, the study of code evolution at the level of individual lines can be extended both theoretically and in empirical breadth and depth. On the theoretical side, significant work is required to establish the precise mechanisms underlying the observed hazard rate. Features we did not examine here, such as as the interplay of requirements, architecture, and syntax, might be worthy candidates for further study. Corresponding theories should then be empirically evaluated using our methods and tools. On the breadth side examining more, and more diverse, repositories will strengthen the generalizability of our findings.

Tying together this area's research and practical implications is the enticing quest to identify and control factors that *do* play a role in the lifetime of code elements. Once these are nailed down, software engineering practices can be correspondingly adjusted so as to reduce potentially wasteful effort by delivering code lines with longer lifespans. This line of research can lead to a new promising and exciting avenue for improving the efficiency of software development.

## ACKNOWLEDGEMENTS

The authors thank Alexander Chatzigeorgiou for his valuable and timely feedback. This work's first author thanks Michiel van Genuchten and Les Hatton for their fruitful collaboration on software growth modeling.

### Funding

The project associated with this work has received funding from the European Union's Horizon 2020 research and innovation program under grant agreement No. 825328.

The funders had no role in study design, data collection and analysis, decision to publish, or preparation of the manuscript.

### Grant Disclosures
The following grant information was disclosed by the authors:
European Union's Horizon: 825328.

### Competing Interests
The authors declare that they have no competing interests.

### Author Contributions
- Diomidis Spinellis conceived and designed the experiments, performed the experiments, analyzed the data, performed the computation work, prepared figures and/or tables, authored or reviewed drafts of the paper, and approved the final draft.
- Panos Louridas analyzed the data, performed the computation work, prepared figures and/or tables, authored or reviewed drafts of the paper, and approved the final draft.
- Maria Kechagia analyzed the data, authored or reviewed drafts of the paper, and approved the final draft.

### Data Availability
Data and source code are available on Zenodo: DOI 10.5281/zenodo.4319993.

- Spinellis, Diomidis, Louridas, Panos, & Kechagia, Maria. (2020). Evolution of software code at the level of fine-grained elements: data files (Version 1.3) [Data set]. Zenodo. DOI 10.5281/zenodo.4319986.

- Kechagia, Maria, Louridas, Panos, & Kechagia, Maria. (2020, December 13). Evolution of software code at the level of fine-grained elements: source code. Zenodo. DOI 10.5281/zenodo.4319993.

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
