# Peer review of "Software evolution: the lifetime of fine-grained elements"

_PeerJ Computer Science, doi:10.7717/peerj-cs.372_

## Round 0.1 · original submission · Minor Revisions

As you can see from the reviews, both reviewers found the paper to already be in very good state. Only minor revisions are required to accept this paper:

- Reviewer 1 sees potential improvements in the replication package. I can see that this may be more work than what's worth it, but I encourage the authors to at least consider refactoring / splitting up the replication package a little (66 GB is indeed a lot of data for us people with an SSD ;) ).

- Reviewer 2 primarily has suggestions for additional literature. I particularly find the suggestion valuable to discuss the usage of the diff algorithm in the context of Nugroho et al. 2020. I am aware that this paper was likely not yet published when the authors have conducted the work (so this isn't a criticism in any way), but now that the paper is out it would be good to consider it in the discussion. I will leave it to the authors to decide to what extent the other literature recommended by the reviewer has to be considered.

Overall, this is a great paper! Congratulations, and I am looking forward to the revised version.

·

Basic reporting

The paper is very well written. Its English is professional, clear and unambiguous.

The academic literature references give enough background and context of the problematic and the context of the research.

The structure (intro, methods, results and discussion, threats, related work and conclusions) is solid, figures and tables are clear.

The raw data and the software are shared in a replication package. As by now, the replication package is provided in a single 3.6GB compressed file (that uncompresses to 66GB). Given that it contains several elements, i.e., the data and the tool, I think it would be a good idea to offer them separately, esp. when 99% corresponds to the repos as I see and the rest to the scripts and stats. The README describes the stats, but the tool (i.e., the many scripts) under the src directory are not documented - I would ask the authors to do it.

The paper is self-contained, with relevant results to the hypotheses presented. At this point it is mainly quantitative, with few qualitative analysis, but I understand that this is to some extent outside of the scope of this work.

Experimental design

The paper addresses an original research, that is within the aims and scope of the journal.

The research questions have been well designed, are relevant and meaningful. The authors elaborate conveniently how their work fills a knowledge gap in the software maintenance and evolution research field.

The research methodology is sound, and it has been performed with high technical and ethical standards. The research shown requires a major non-trivial engineering effort (e.g., the token-based approach in addition to just considering lines, or the git branch flattening), that is presented correctly in the paper and that is available to other researchers in the replication package. I've already pointed out some limitations in the replication package that I would the authors to address.

The methods and sources used are described with detail in order to make a replication possible. In case other researchers want to replicate the study, the replication package is available and contains even the analyzed repositories.

Validity of the findings

All underlying data have been provided. The findings are robust, statistically sound, and controlled.

Conclusions are well stated, linked to the original research question and limited to supporting results.

The research is mainly quantitative, so there is speculation on the reason for the results. However, the speculative part -some of the discussion points- is small, and can be clearly identified even if it shares section with results. I understand that this is a task for further research, and not necessarily has to be included in this work, which already contributes significantly with new scientific knowledge.

Additional comments

The inclusion and removal of large amounts of external code has been reported previously in software evolution studies. My feeling is that they affect severely the results obtained, as it is the case for Handbrake. This is mostly external code and may be considered differently (e.g., in some software growth studies some authors filter these ripples out). I would ask the authors to elaborate a little bit more about it -- at least, it should be included as one of the threats to validity.


Other, minor issues:

Line 99: constructs. drivers of change.

Line 252: copied to renamed --> copied or renamed?

Line 328: if we have line --> if we have a line

Lines 503, 549, 617: Section )

Line 516: Sections 6 and 6

Line 558: All lines of code in these projects die young -> Really they do not die... they are still there, maybe in a zombie state as the project is discontinued, but have not been removed.

·

Basic reporting

Regarding the literature references of software evolution, although the paper describes macroscopic and microscopic studies, I think the paper misses the discussion of other elements of source code like methods:
- Jennifer Bevan, E. James Whitehead, Sunghun Kim, and Michael Godfrey. 2005. Facilitating software evolution research with kenyon. In Proceedings of the 10th European software engineering conference held jointly with 13th ACM SIGSOFT international symposium on Foundations of software engineering (ESEC/FSE-13).
- Thomas Zimmermann. 2006. Fine-grained processing of CVS archives with APFEL. In Proceedings of the 2006 OOPSLA workshop on eclipse technology eXchange (eclipse '06).
- Hideaki Hata, Osamu Mizuno, and Tohru Kikuno. 2011. Historage: fine-grained version control system for Java. In Proceedings of the 12th International Workshop on Principles of Software Evolution and the 7th annual ERCIM Workshop on Software Evolution (IWPSE-EVOL '11).

The idea of creating a synthetic Git repository where files are stored as the targeted elements was originally presented in the paper:
- Hideaki Hata, Osamu Mizuno, and Tohru Kikuno. 2011. Historage: fine-grained version control system for Java. In Proceedings of the 12th International Workshop on Principles of Software Evolution and the 7th annual ERCIM Workshop on Software Evolution (IWPSE-EVOL '11).

Minor:
In the section of Internal Validity and External Validity, there are some minor issues:
- No section name in "(Section )"
- "Section 6 and 6"

Experimental design

Regarding the diff used in this study, a recent study reported that among the diff algorithms built into Git, Histogram is better than Myers:
Nugroho, Y.S., Hata, H. & Matsumoto, K. How different are different diff algorithms in Git?. Empir Software Eng 25, 790–823 (2020).
In light of this finding, please discuss the problems with using traditional diff in this study.

As for the death of lines, I think line death would also occur by deleting the file to which the line belongs. I could not understand how to measure the lifetime of lines when files are removed. Please explain in such a case.

Validity of the findings

Regarding the tool lifetime, although it is included in the replication package, I think it is valuable to be presented as a separate project in GitHub.

Additional comments

This paper studies the evolution of source code lines and tokens in 89 Git repositories. To conduct this study, the authors presented methods of Git history simplification, survival tracking, token history analysis, and analysis of moved lines. Standard parametric models in survival analysis were built to model the lifetime of source code lines and tokens.

The description of the method is almost complete, and the publication of the tools and dataset is a valuable contribution. There were just a few manageable problems, but they can be mostly solved by adding the description.

---

## Round 0.2 · accepted · Accept

Thank you for your submission to PeerJ CS!

I see no reason to further delay the publication of this paper, and will mark it as an Accept now.